



# Characterizing Soil Moisture Drought Onset and Termination in Europe

Woon Mi Kim[1,4] and Santos J. González-Rojí[2,3,5]

[1]Climate and Global Dynamics Laboratory, NSF National Center for Atmospheric Research, Boulder CO, United States
[2]Climate and Environmental Physics, University of Bern, Bern, Switzerland
[3]Oeschger Centre for Climate Change Research, University of Bern, Bern, Switzerland
[4]now at: Institute of Meteorology and Climate Research Troposphere Research (IMKTRO), Karlsruhe Institute of Technology, Eggenstein-Leopoldshafen, Germany
[5]now at: Department of Physics, University of the Basque Country (UPV/EHU), Leioa, Spain

**Correspondence:** Woon Mi Kim (woon.kim@kit.edu)

**Abstract.** Despite significant advances in understanding the mechanisms and drivers of droughts, climatological characteristics of soil moisture drought onset and termination (DO&T) in Europe remain relatively under-explored. Therefore, this study investigates the temporal characteristics of soil moisture DO&T and their connections to large-scale circulation patterns in Europe during 1980–2020. We analyze the duration of transition to DO&T, their seasonality, and atmospheric circulation
patterns associated with each drought phase. The regions of study are central (CEU) and Mediterranean (MED) Europe, using soil moisture datasets from ERA5-Land, GLEAM version 3, SoMo.ml, Noah-LSM, and a simulation from the Community Land Model version 5 (CLM-TRENDY).

Our findings indicate that the duration of transition to DO&T depends on the definition of the transition period and the soil moisture dataset used. When the transition period is defined based on precipitation, the onset is longer than the termination.
When the comparison is performed among the datasets, GLEAM shows the longer transition to DO&T, in contrast to ERA5-Land, which exhibits a shorter transition to DO&T. These discrepancies indicate that the selection of definitions and datasets can influence drought-related decision-making. Furthermore, we observed shifts in drought development speed during the study period: onset has become faster while termination has become slower in the eastern Mediterranean region.

In terms of seasonality, onset occurs more frequently during the wet seasons – summer and autumn in CEU and autumn and
winter in MED – while termination is less seasonally constrained and shows more discrepancies among the datasets. Lastly, the effects of North Atlantic Oscillation (NAO), Scandinavian, East-Atlantic, and East Atlantic-West Russian (EA-WR) patterns on DO&T are assessed. Different patterns can serve as early warnings of droughts depending on the seasons and regions. For instance, it can be confirmed that NAO is important for the Iberian Peninsula, while EA-WR influences DO&T for all regions during winter. Overall, this characterization of DO&T can provide a reference for evaluating potential future changes
in drought characteristics, which are expected to be altered by global warming.





## 1 Introduction

Drought is a climate phenomenon characterized by a prolonged period of dry conditions (Dai, 2011). When depletion of water occurs in the soil, the event is referred to as a soil moisture drought. Soil moisture droughts can result from a persistent dry period with little precipitation (meteorological drought) alone or can arise from the combined influence of reduced precipitation and increased evapotranspiration. The impacts of soil moisture droughts can be far-reaching, affecting ecosystems and a wide range of socioeconomic sectors (Naumann et al., 2015; Bachmair et al., 2016) with clearly apparent impacts on agricultural activities (Moravec et al., 2021). The damage caused by droughts becomes increasingly severe with the duration of the events. Although dry-wet fluctuations are a natural aspect of a region's hydroclimate, there is evidence that anthropogenic warming has perturbed the natural characteristics of droughts through an increase in vapor pressure deficit, the intensification of the global hydrological cycle, and the manipulation of water storage (Trenberth, 2011; Van Loon et al., 2016b; Seneviratne et al., 2021).

In the last decade, Europe has experienced multiple intense droughts, marked by low soil moisture and accompanied by record-breaking summer heatwaves (e.g., Hari et al., 2020; Sousa et al., 2020; Moravec et al., 2021; Rakovec et al., 2022). Various studies have examined the drivers of individual droughts during this period. For example, the intense summer 2015 (Ionita et al., 2017) and 2018 (Moravec et al., 2021) droughts were driven by upper-level ridges and blocking highs, which were associated with a weakening of the subtropical jets. The 2016/2017 drought that affected almost the entire Europe was characterized by consecutive blocking events and subtropical ridges, which in turn weakened zonal circulation and moisture transport from the Atlantic (García-Herrera et al., 2019). During this drought event, the impact of temperature that drove an increase in atmospheric evaporative demand was more pronounced in southern Europe, and a similar situation was found for the 2022 event (Faranda et al., 2023). Vicente-Serrano et al. (2021) have indicated that the contribution of vapor pressure deficit on droughts has significantly increased in western Europe since the mid-20th century.

Climatologically and on monthly and seasonal time scales, soil moisture and drought variability in Europe are influenced by large-scale circulation modes, for instance, the Arctic Oscillation (AO) and the North Atlantic Oscillation (NAO; Markonis et al., 2018; Almendra-Martín et al., 2022) and East Atlantic-West Russia pattern (EA-WR; Kingston et al., 2015). These circulation modes are linked with the intensity of westerlies, which bring moisture from the northern Atlantic to Europe (Vicente-Serrano et al., 2016). NAO, in particular, has lagged effects on monthly soil moisture variability, exhibiting negative correlations across a wide region in Europe (Almendra-Martín et al., 2022). Vicente-Serrano et al. (2011) showed that the effect of winter NAO is propagated to drought conditions in the following months in the Mediterranean, with positive NAO relating to drought conditions. Lhotka et al. (2020) identified atmospheric circulation patterns that are connected to the severity of droughts in central Europe. They observed that increased drought frequency and severity during the vegetation period (April–September) are linked with dry circulation types characterized by anticyclonic conditions and/or dry advection. Wet circulation types are associated with cyclonic circulation, and they show a negative trend in the 1948–2018 vegetation periods. They concluded that increased dry circulation types have significantly contributed to the severity of 2015–2018 dryness in Europe.





While drought intensification and its drivers have been relatively well-studied, the characteristics of drought onset and termination remain comparatively under-explored. This can be because determining the precise timing of initiation and termination of droughts is challenging, primarily due to the complex and multifaceted nature of droughts (Cook et al., 2018). The drivers of drought onset and termination can be diverse, from large-scale climate patterns to regional-scale processes, including synoptic weather systems and land-atmosphere interactions, which are also influenced by the land surface characteristics. Each of these processes has different time scales of influence on soil moisture variability. Another complexity arises from the fact that droughts are typically slow events, whose impacts on the ecosystem become noticeable when the events already reach the mature phases, with noticeably progressed accumulated periods of precipitation or soil moisture losses (Wilhite, 2000; Bachmair et al., 2016). These factors collectively make it challenging to define when exactly a dry or wet condition that initiated or ended a drought has commenced. However, investigating drought phases, particularly onset and termination, is a topic that requires more attention to understand better the mechanisms of droughts and improve early predictions of such events.

Several studies that investigated a life cycle of droughts have defined drought phases based on different standardized drought indices, accumulated precipitation, or soil moisture anomalies (e.g., Mo, 2011; Seager et al., 2019; Shah and Mishra, 2020; Řehoř et al., 2021). Although the criteria for defining drought onset and termination differ across the studies, in general, a drought onset is the time when a drought index falls below a threshold, and a drought termination is a recovery period from the minimum water balance to a normal condition. Typical duration of drought phases and drivers involved during these periods have also been investigated. For Europe, Řehoř et al. (2021) examined atmospheric circulations associated with the three phases of soil moisture droughts in the Czech and Slovak Republics during 1961–2019. Using the datasets from the SoilClim model, they found that anticyclonic circulation types linked with low precipitation occur more frequently during the initiation (onset) and throughout droughts. The opposite cyclonic circulation types that bring precipitation to the region are more frequently observed during the recovery phase of droughts. Parry et al. (2016) reviewed drought terminations in the British Isles, indicating different synoptic events that affected stream flow and terminated droughts. Margariti et al. (2019) investigated anthropogenic influences on droughts in Europe in the present day, demonstrating that human activities have extended the termination of streamflow droughts.

Despite these significant advances in the understanding of droughts and drivers of the individual events presented above, the climatological characteristics of soil moisture drought onsets and terminations in Europe are still a relatively under-explored topic. Climatological analyses of droughts have been conducted using standardized drought indices (e.g., Lloyd-Hughes and Saunders, 2002; Spinoni et al., 2015), but they did not focus on the characteristics of each of the drought phases. Future changes are expected to affect temporal characteristics of droughts (Trenberth, 2011; Cook et al., 2018), potentially inducing changes in drought onset and termination (Walker and Van Loon, 2023; Yuan et al., 2023). To assess the potential changes in droughts, it is clearly necessary to perform a climatological characterization of drought onset and termination in Europe, which will also aid in enhancing early predictions and readiness for such extreme events.

The objective of this study is to investigate temporal characteristics of onset and termination of soil moisture droughts and connections to large-scale circulation patterns in Europe during 1980–2020. First, we examine the climatological duration, trends, and seasonality of drought onset and termination. Then, we examine the connection between drought onset and termi-





nation and large-scale circulation modes in the North Atlantic domain. The primary study regions are central and Mediterranean Europe, which are the regions in Europe that have been most affected by droughts in the last decade and where soil dryness is expected to increase in future warming scenarios, with medium (in central Europe) and high (in southern Europe) confidence levels (Seneviratne et al., 2021). We use five soil moisture datasets from different sources. As most of the previous literature has employed only one dataset from a single project, by using several soil moisture datasets from diverse projects we aim to examine the robustness of the temporal characteristics of drought. The description of the datasets is given in Section 2. The definition of drought phases and other methods employed in the analysis are introduced in Section 3. All the results on the climatological duration, trends, seasonality of onsets and terminations, and the relationships of drought onset and termination and large-scale circulation patterns are provided in Section 4. Finally, we present the discussion (Section 5) and conclusion in (Section 6).

## 2  Data

### 2.1  Soil moisture datasets

We use five gridded soil moisture products: three are the outputs of land surface models (LSM) forced by observation-based meteorological fields, one is based on a multi-layer soil model and assimilated with satellite-based products, and the last one is a machine-learning-trained observation-based dataset. The three soil moisture datasets from LSMs are ERA5-Land (Muñoz-Sabater et al., 2021), Noah-LSM (Koren et al., 1999) from the Global Land Data Assimilation System project (GLDAS; Rodell et al., 2004), and TRENDY (Friedlingstein et al., 2022) from the Community Land Model version 5 (CLM5; Lawrence et al., 2019). The next dataset is Global Land Evaporation Amsterdam Model soil moisture (GLEAM; Miralles et al., 2011; Martens et al., 2017), and the last one is SoMo.ml, which is derived from a machine learning-based model (O and Orth, 2021). Where available, evapotranspiration is also retrieved from these datasets.

We mostly rely on outputs from gridded datasets because our study requires continuous soil moisture data to identify drought phases properly, and many solely observation-based soil moisture datasets are generally short and not continuous in time and space. The datasets are summarized in Table 1, and a brief description of each dataset follows.

ERA5-Land (Muñoz-Sabater et al., 2021) uses the offline ERA5 Land Surface Model, and it is forced by the atmospheric variables from the ERA5 reanalysis. The land processes are based on the ECMWF Scheme for Surface Exchanges over Land with land surface hydrology from the H-TESSEL model. The horizontal resolution is $0.25° \times 0.25°$, and we employ the data with the monthly time resolution.

GLEAM version 3 (Miralles et al., 2011; Martens et al., 2017) is a set of algorithms to estimate global evaporation that aims to provide an advanced representation of evaporation based on satellite and reanalysis forcing. The dataset contains not only terrestrial evaporation but also surface and root-zone soil moisture. Soil moisture is estimated using a multi-layer running-water balance model, and the upper-level soil moisture is assimilated with satellite-based microwave soil moisture. The spatial resolution of the dataset is $0.25° \times 0.25°$ covering the period 1980 to the present.



SoMo.ml v1 (SoMo; O and Orth, 2021) is a global daily soil moisture dataset reconstructed through a machine learning model trained with in-situ soil moisture measurements across the globe. SoMo employs a Long Short-Term Memory neural network to reconstruct the daily global soil moisture field. The predictors fed into the model are the meteorological variables from reanalysis and remote sensing datasets, and the target variable is soil moisture from 1000 in-situ measurements across

the globe. The means and standard deviations of the daily in-situ soil moisture are scaled up to match those of the ERA5 grid cells to produce seamless merging across different stations and time series. The spatial resolution of SoMo is 0.25°, and the daily temporal resolution is converted to the monthly values by calculating the monthly averages. The temporal coverage of the dataset is from 2000 to 2019.

GLDAS (Rodell et al., 2004) is a project comprising various land surface models that provide global land surface variables.

The output from GLDAS version 2.1 is used taking the period 2000–2020. The atmospheric input forcing for GLDAS combines data from models and observations, including the Princeton meteorological forcing (Sheffield et al., 2006). From the LSMs that comprise GLDAS, we use Noah LSM (Noah; Koren et al., 1999) that provides the surface level soil moisture. We employ the dataset with the spatial resolution of $1° \times 1°$ and the monthly average temporal resolution.

TRENDY (CLM-TRENDY) is a simulation from the offline CLM5 (Lawrence et al., 2019) and is part of the Global Carbon

Budget 2022 project (Friedlingstein et al., 2022). The simulation was performed with a transient $CO_2$ and land use change from 1701 to 2021 and forced by the merged Climate Research Unit (CRU) – Japanese 55-year Reanalysis (JRA55) atmospheric forcing from 1901 onward. Before 1901, the atmospheric forcing during 1901–1920 is cycled over to fill the period. The dataset has a spatial resolution of approximately $0.95° \times 1.25°$ and a monthly temporal resolution.

For precipitation ($P$), the forcing data corresponding to each LSM mentioned above are utilized. For SoMo, we use ERA5-

140 Land P, and for GLEAM, whose forced merged precipitation data is not explicitly provided, we employ P from E-OBS. E-OBS (version 27.0e; Cornes et al., 2018), is a gridded daily precipitation dataset over Europe (land-only, 25°N–71.5°N, 25°W–45°E) and is based on weather station data collected by the ECA&D initiative (Klein Tank et al., 2002; Klok and Klein Tank, 2009). E-OBS has a spatial resolution of $0.1° \times 0.1°$, and is available from 1950 onward. We evaluated that the spatial variability in monthly P is minimal between the datasets, with consistent spatial means and annual cycles (to be discussed later

in Fig. 7).

## 2.2 Large-scale climate modes

Four modes of circulation patterns that are known to influence the climate in the Euro-Atlantic region are taken for the analysis: the North Atlantic Oscillation (NAO; Hurrell, 1995; Hurrell et al., 2003), and the Scandinavian (SCA; Barnston and Livezey, 1987; Bueh and Nakamura, 2007), East Atlantic (EA; Wallace and Gutzler, 1981; Comas-Bru and Hernández, 2018), and East

Atlantic-West Russian (EA-WR; Barnston and Livezey, 1987; Lim, 2015) patterns. The indices that correspond to each pattern are all on monthly temporal resolutions and retrieved from the NCAR climate data guide (https://climatedataguide.ucar.edu/type/climate-indices/circulation/nao Schneider et al., 2013) for NAO and the KNMI climate explorer (https://climexp.knmi.nl/) for the others.





**Table 1.** Soil moisture datasets employed in this study.

| Dataset name (Abbreviation) | Institution | Type | Temporal resolution | Spatial resolution | Reference |
|---|---|---|---|---|---|
| ERA5-Land (ERA5-Land) | ECMWF | Offline LSM | 1950–Present | 0.25° × 0.25° | Muñoz-Sabater et al. (2021) |
| GLEAM v3 (GLEAM) | ESA | Multi-layer soil model and assimilated | 1980–2021 | 0.25° × 0.25° | Miralles et al. (2011) Martens et al. (2017) |
| SoMo.ml (SoMo) | MPI Biogeochemistry | Machine-learning trained model | 2000–2019 | 0.25° × 0.25° | O and Orth (2021) |
| Noah-LSM from GLDAS v2.1 (Noah) | NASA | Offline LSM | 2000–Present | 1° × 1° | Rodell et al. (2004) |
| CLM-TRENDY (CLM-TRENDY) | NCAR | Offline LSM | 1701–2021 | 0.95° × 1.25° | Lawrence et al. (2019) Friedlingstein et al. (2022) |

NAO represents the fluctuation between the Icelandic low and the Azores high, and the index is calculated as the leading
modes of Empirical Orthogonal Function (EOF) of the sea level pressure anomalies over the Atlantic sector, encompassing
20°–80°N and 90°W–40°E. SCA, also referred to as the Eurasian Type 1 pattern, is a pressure system with a circulation center
over the Scandinavian Peninsula and other two weaker centers of opposite signs in the northeastern Atlantic and Mongolia. EA
refers to a north-south dipole of pressure anomaly centers in the North Atlantic and Europe. Due to its north-south dipole of
anomaly centers, it is structurally similar to NAO, therefore, considered a southward displaced NAO pattern. Lastly, EA-WR,
also known as the Eurasian Type 2 pattern, is characterized by pressure anomaly centers located over the Caspian Sea and
western Europe. The indices for SCA, EA and EA/WR are the first three or four leading modes of the rotated EOF of the
monthly mean geopotential height fields over the North Atlantic and Europe.

## 3 Methods

### 3.1 Region of study

We focus on Europe, emphasizing more central (CEU) and Mediterranean Europe (MED). These regions are confined approxi-
mately between 45°N–56.5°N and 9.5°W–25.5°E for CEU and between 36.5°N–45°N and 9.5°W–25.5°E for MED following
the division based on the IPCC climate reference (Iturbide et al., 2020).

Additionally, we divide CEU and MED into more subdivisions to consider different regional mean hydroclimate conditions
before assessing preferred seasons for drought onset and termination. CEU and MED are then separated into six, following sim-
170 ilar divisions employed by Christensen and Christensen (2007) (Fig. 1): Iberian Peninsula (IP), east-southern Europe (EMED),
Alps (ALP), France (FR), mid-central Europe (MCEU), and east-central Europe (ECEU).



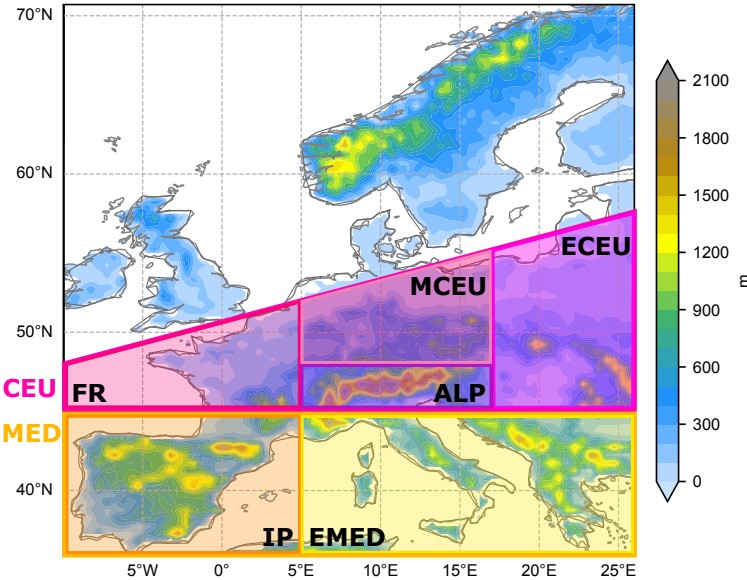

**Figure 1.** Topography in Europe and regional divisions based on Christensen and Christensen (2007) and Iturbide et al. (2020).

## 3.2 Metrics for soil moisture droughts

We use surface-level soil moisture, which is at the upper 10 cm. This layer is chosen as it is the most commonly available among the datasets. GLEAM, SoMo, Noah, and CLM-TRENDY directly output 10 cm-soil moisture. As this level is not available in

ERA5-Land, the corresponding value is obtained by estimating how much soil moisture is in the first 3 cm of the second layer (7–28 cm), then adding this amount onto the soil moisture of the first layer (7 cm). The assumption is that there is no gradient of soil moisture within the second layer.

Although the surface layer soil moisture cannot indicate dryness occurring in deeper soil layers, this depth is where soil moisture can be measured over a broad spatial extent and is commonly available in most soil moisture products. Moreover,

surface layer soil moisture is highly correlated to the widely used operational drought indices, for instance, the Standardized Precipitation Evapotranspiration Index (SPEI; Vicente-Serrano et al., 2009). Additionally, the variable is directly affected by changes in atmospheric circulation; hence, it is suitable for examining the influences of large-scale climate drivers on soil moisture.

The drought index to quantify soil moisture droughts is the three-month running mean of standardized surface soil moisture

anomalies ($SM3$), same as Mo (2011). Running averages over three months smooth out noises in the monthly time series by considering the typical seasonal time scale in the mid-latitudes. To obtain SM3, first, the anomalies of soil moisture ($SM$) are calculated at each grid point over the study region. The time series of soil moisture are deseasonalized through subtraction of the 2000–2014 (15-year) annual cycles, and then standardized using the multi-year standard deviations corresponding to each month. These 15 years are chosen for standardization as it is the common period to all five datasets and excludes the intense





soil moisture dryness in CEU that began in 2015. The three-month running means are calculated using these standardized soil moisture anomalies. The SM3 time series have a monthly time resolution, where an individual time step $t$ encompasses the soil moisture conditions over the previous three cumulative months, from $t_{-2}$ to $t_0$. For instance, the anomaly of February 1981 indicates the soil moisture conditions during December 1980–February 1981. This approach of defining a drought index with a non-centralized moving average has an operational purpose, making it suitable for monitoring drought conditions for the

current time step. Additionally, this method aligns with other existing drought indices such as the Standardized Precipitation Index (SPI) (McKee et al., 1993) and SPEI.

In addition, we calculate monthly P anomalies, $\Delta P$ calculated by subtracting the 2000–2014 annual cycle from P at each grid point.

### 3.3    Definitions of drought onset, termination, and the transition periods

A drought is a period when SM3 stays below one negative standard deviation (-1$\sigma$). A threshold of -1$\sigma$ in a standardized drought index corresponds approximately to the 15.9th percentile level, and based on the classification of drought categories for other standardized drought indices (i.e., SPI or SPEI; Lloyd-Hughes Benjamin and Saunders Mark A., 2002), values below this threshold indicate moderate to extreme droughts.

A drought onset (DO) is the first monthly time step that SM3 reaches below -1$\sigma$ after a continuous period of positive SM3

(Fig. 2). On the contrary, a drought termination (DT) is the first monthly time step that SM3 reaches above -1$\sigma$ from the minimum SM3. A drought is fully terminated when SM3 exceeds 0 for two consecutive time steps. This temporal criterion is to ensure a full recovery from drought. A transition period to DO&T is estimated using two approaches: the first approach is fully based on the same SM3 index, similar to Seager et al. (2019), and the second approach uses monthly precipitation anomalies ($\Delta P$), following Mo (2011) and similar to Řehoř et al. (2021).

The transition periods to DO&T based on SM3 is illustrated in Fig. 2a and defined as follows:

- A transition period to DO consists of consecutive months with negative SM3, from the beginning of the dry period (SM3 $\leq$ 0) until DO, which is the first instance when SM3 crosses -1$\sigma$ in the direction of negative SM3. If a drought begins rapidly with SM3 falling below -1$\sigma$ without any intermediate month, the transition period to DO is one month long, consisting of only the month of DO.

- A transition period to DT is the number of months from the minimum SM3 within a drought until DT, which is the first time SM3 crosses -1$\sigma$ in the direction of positive SM. For this, a time step of minimum SM3 needs to be calculated beforehand, which is the period with the maximum dryness. This definition of termination follows Parry et al. (2016). As same to DO, a transition period to DT is one month long when the drought ends rapidly without any transition months.

The transition period to DO&T based on $\Delta P$ is seen in Fig. 2b, which is described as:

- A transition period to DO is defined as the months with negative $\Delta P$ within the 12 months before reaching DO. This reflects the cumulative dry period until its effect on soil moisture becomes noticeable. The 12-month scale is chosen




because it is the typical time scale of drought indices, i.e., SPEI and SPI (Mo, 2011). Negative $\Delta P$ in the transition period do not need to be consecutive in time.

– Similar to the transition period to DO, the transition period to DT is the months with positive $\Delta P$ within the 12 months preceding DT. Therefore, it refers to a cumulative wet period necessary to fill soil moisture to end drought.

To estimate the transition periods, we use the 12-month time scale for $\Delta P$ used by Mo (2011) to allow comparison with their result. However, we also analyze the transition periods using a 6-month time scale to assess whether the duration of the DO&T transition periods depends on the chosen time scale for $\Delta P$ accumulation. By applying two different definitions of transition periods adapted from various studies, we evaluate the sensitivity of the duration of the DO&T transition periods to
the definitions.

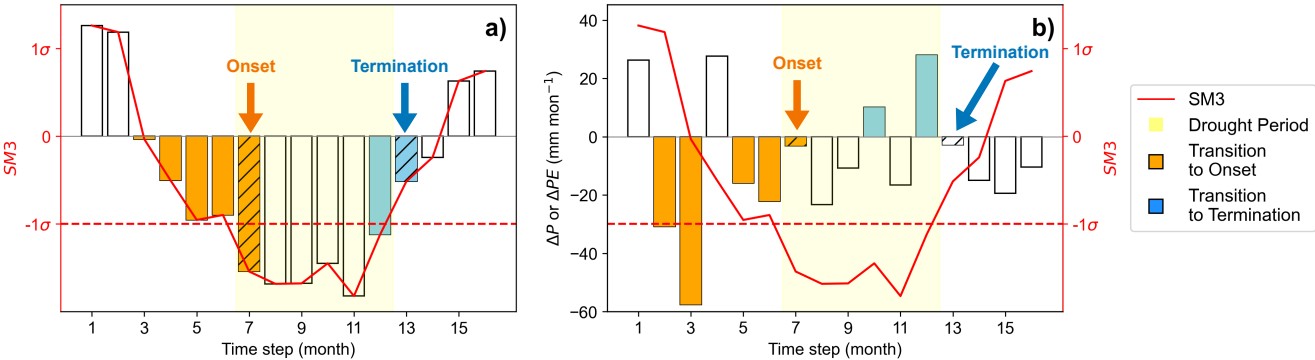

**Figure 2.** Illustration of drought onset (DO) and termination (DT) a) based on soil moisture anomaly (SM), and b) based on precipitation anomaly ($\Delta P$). The red curves in (a) and (b) are the same and indicate SM and the red dashed lines are the $-1\sigma$ of SM for the drought threshold.

### 3.4    Duration of DO&T, trends, and their preferred season of occurrence

We calculate the duration of the transition period to DO&T – hereafter, simply referring to the DO&T duration – at each grid point over the study region in all five soil moisture datasets. Statistical assessment to compare the mean duration between DO and DT within the same dataset and the mean duration of DO or DT between the datasets are performed using t-tests (Wilks,
2011). In the t-tests, the null hypothesis assumes that the means of the two phases or two datasets are derived from the same population. Our test, therefore, assesses whether their means are being sampled from the same population at a 95% level using a two-sided test. We perform the statistical tests for each region, CEU and MED.

Next, we evaluate whether the DO&T duration has changed during the study period by calculating linear trends of the DO&T duration at each grid point. We perform this analysis in three soil moisture datasets, ERA5-Land, GLEAM, and CLM-
TRENDY, since they cover a longer period (1980–2020), and we can include more droughts and their duration. The significance





of the slopes is tested using a two-sided Wald test at a 95% confidence level, where the null hypothesis states that the slope is zero.

Using the same three datasets, we estimate the preferred seasons for DO&T. Since the study regions are located in the mid-latitudes, a standard division into the four boreal seasons is adopted (DJF is winter, MAM is spring, JJA is summer, and SON

is fall). A DO or DT is said to occur in a given season if the time step of DO or DT falls within that given season. The number of DO&T within a season is counted at each grid point, and then, the ratio between the number of DO (or DT) occurrences during a given season and the total DO (or DT) occurrences is calculated. If droughts occur uniformly across the four seasons, the occurrence ratio would be 0.25 for each season.

## 3.5   Connection of large-scale circulation patterns to DO&T

We address the connections between the transition to DO&T and large-scale climate modes, defined in Section 2.2, by counting the frequency of each mode during the transition phases – either to DO or DT. This allows us to analyze whether climate modes exhibit distinct frequencies and mean values preceding DO and DT. The statistical significance of the climate modes associated with each transition phase is assessed using a t-test at a 95% confidence level, comparing the values against the climatological means.

## 4   Results

### 4.1   Duration and trends of transition to drought onset and termination

The duration of DO&T is estimated for the period 2000–2019 for SoMo and 2000–2020 for the other four soil moisture datasets. The values are presented in Fig. 3 for DO and Fig. 4 for DT, including the spatial averages over CEU and MED regions provided in Table 2. Additionally, the mean duration for the entire period (1980–2020) for ERA5-Land, GLEAM, and

CLM-TRENDY datasets can be found in the supplementary Figures S1 and S2.

Figs. 3 and 4 show that the duration of both DO and DT is longer in the P12-based method compared to the SM3-based DO&T. This difference demonstrates a clear dependence of DO&T duration on the definition of the transition period. A longer P12 period signifies that a relatively longer period of precipitation deficit or accumulation is required until the effect of precipitation anomalies is reflected in soil moisture to induce DO or DT.

Another observation is that there are clear differences in the duration of DO&T among the datasets. In general, GLEAM exhibits a longer DO&T duration in both P12 and SM3, while ERA5-Land shows the shortest duration compared to the other four datasets. This discrepancy is also consistent with the duration of droughts (not shown), with GLEAM presenting droughts with longer duration than other datasets, while on the contrary, ERA5-Land shows droughts with relatively shorter duration. The difference in drought duration is reflected in the number of droughts: GLEAM has droughts with longer duration, but it

presents fewer events (number of events $N$=6) and the opposite for ERA5-Land (N=10; Figs. 3 and 4).



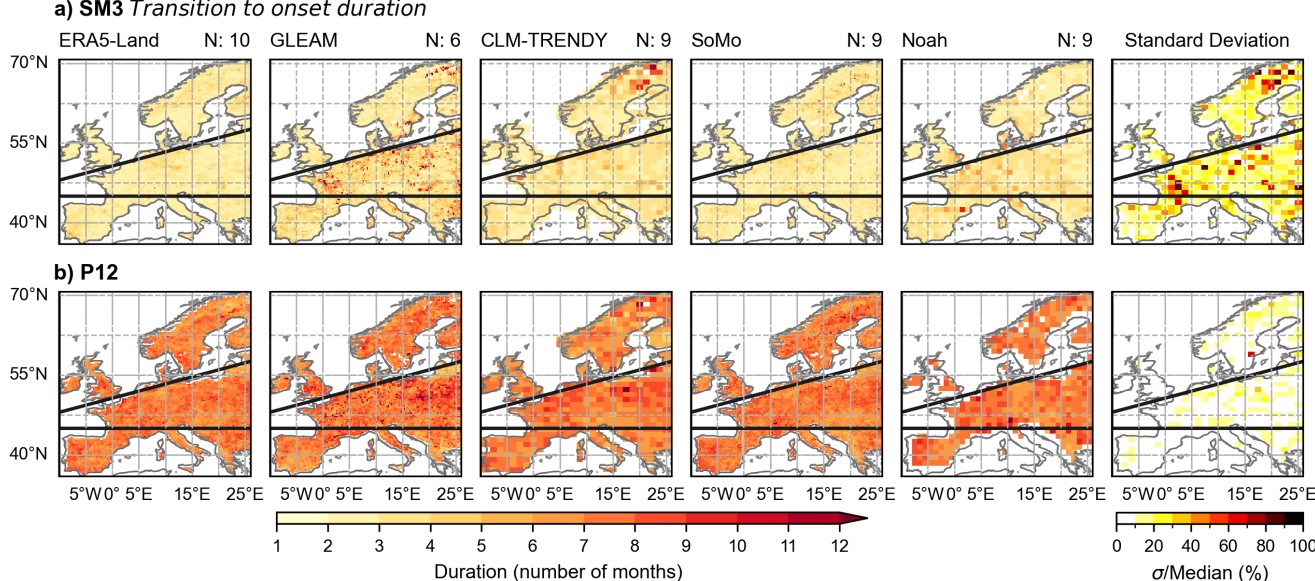

**Figure 3.** The mean DO duration (drought onset) during 2000–2019 for SoMo and 2000–2020 for the other four datasets calculated based on a) SM and b) P12 (Section 3.2a.). N at the top right part of each panel indicates the median of the number of drought events over CEU and MED. The standard deviations of the mean duration across the five soil moisture datasets are shown in the rightmost column. Black thick lines separate central Europe (CEU) and Mediterranean Europe (MED).

When the comparison is performed in the duration between the drought phases, namely DO and DT, within the same dataset, the difference between DO and DT appears to be smaller in SM3 than in P12. In P12, DO exhibits a significantly longer duration than DT over almost the entire Europe. This suggests that a more prolonged cumulative period of precipitation deficits is needed to lead to soil moisture depletion to initiate droughts compared to the cumulative positive precipitation anomalies
required to terminate droughts. A longer transition period to DO than DT is a finding that is consistent with Mo (2011) for the U.S. droughts. For SM3, the difference between DO and DT is not pronounced and varies with the datasets, with GLEAM showing a slightly larger difference between DO and DT.

There are also spatial variations in the duration, regardless of the datasets, drought phases, and transition definition. Some regions exhibit longer DO&T duration than others, with sporadic values distributed across the space. In general, CEU tends
to show slightly longer DO&T duration than MED, in which this difference is more pronounced in the DT (Fig. 4) and in GLEAM particularly.

Table 2 summarizes DO&T duration averaged over each CEU and MED, and it reiterates the observations described above. The DO&T duration based on SM3 is much shorter than that based on P12 for both CEU and MED. The t-tests indicate that, for most of the datasets and regions, DO and DT are statistically different to each other. In the SM3-based duration, DT is
285 generally slightly longer than DO for both regions (the maximum difference is 1.56 months), while in the P12-based duration,



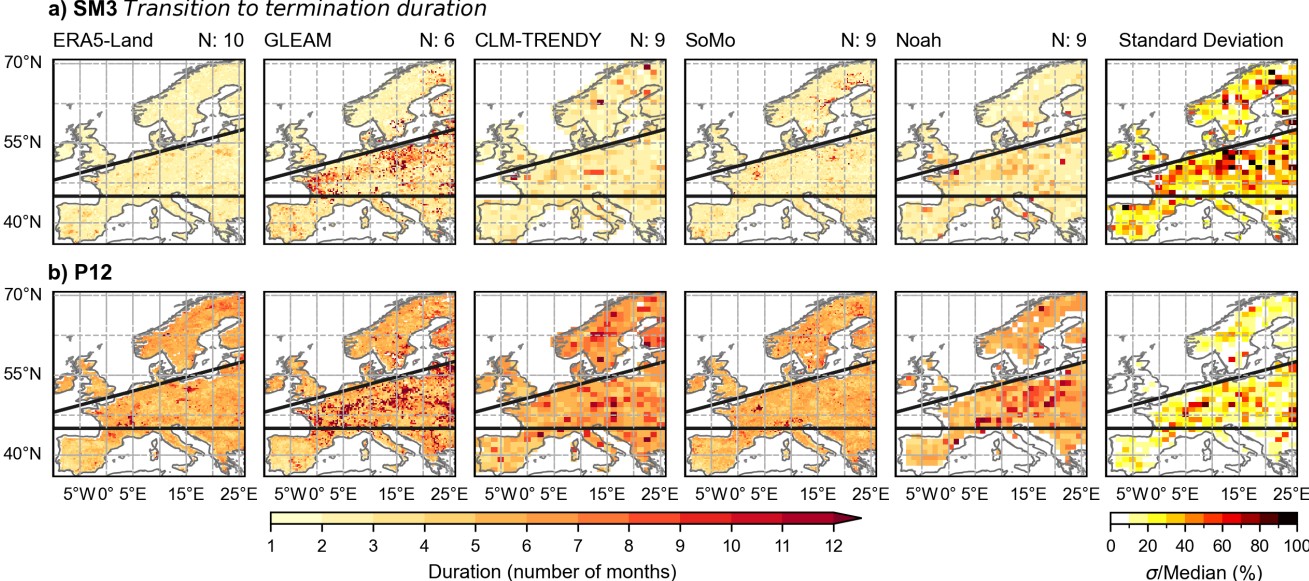

**Figure 4.** Same as Fig. 3 but for DT.

DO is longer than DT (the maximum difference is 1.83 months). The differences in the DO&T duration between the two regions, CEU and MED, within the same dataset are not as pronounced as the differences between the transition definitions, i.e., SM3 or P12, the datasets, and the drought phases. Again, GLEAM shows the higher averaged SM3-based DO&T duration, although not in the P12-based duration. This result highlights the sensitivity of the DO&T duration to the chosen transition definition. When we repeated the same analysis with P6 (Supplementary Figure S3), we obtained a similar result, indicating the dependence of DO&T duration on the dataset and the transition definition.

Next, we assess whether the transition periods to DO&T present trends during the study period, 1980–2020. Fig. 5 presents the linear trend coefficients of the P12-based DO&T duration in ERA5-Land, GLEAM, and CLM-TRENDY. In general, the majority of the areas indicate statistically non-significant changes in the duration of DO&T over time, with very small coefficient values for both DO&T. Nevertheless, it is noted that the majority of the statistically significant trends, which are around 7% to 12% of the grid points, are located over the eastern Mediterranean, encompassing Greece and the Balkan area in all three datasets.

These regions exhibit negative trends in the DO duration (Fig. 5a), which implies a decrease in the transition period to DO. On the contrary, the region presents positive trends for the transition to DT (Fig. 5b), indicating an increase in drought termination periods. Statistically negative trends in the DO duration mean that DO has become faster during 1980–2020, while significant positive trends in the DT duration suggest that the time required to terminate droughts has become longer. These changes in the transition periods seem to indicate potential changes in temporal characteristics of drought, which is further discussed in Section 5.





**Table 2.** Averaged duration in months of the transition periods to DO&T (simply referred to as the DO&T duration) from Figs. 3 and 4 for CEU and MED. When the DO and DT duration within the same dataset are statistically different from each other based on the t-tests at a 95% confidence level, the values are denoted with *. When the DO (or DT) of a dataset are statistically different from DO (or DT) of one or some of the datasets based on the t-test, they are denoted with **.

| | CEU | | | | MED | | | |
|---|---|---|---|---|---|---|---|---|
| | SM3 | | P12 | | SM3 | | P12 | |
| | DO | DT | DO | DT | DO | DT | DO | DT |
| **ERA5-Land** | 2.91*,** | 3.22*,** | 6.89* | 5.33*,** | 2.98** | 2.96** | 6.91* | 4.80*,** |
| **GLEAM** | 3.99*,** | 5.55*,** | 7.21*,** | 6.52*,** | 3.69*,** | 3.90* | 6.74* | 4.94* |
| **CLM-TRENDY** | 3.36*,** | 3.59*,** | 7.02* | 5.93* | 3.23** | 3.37** | 6.77** | 5.37*,** |
| **SoMo** | 3.02*,** | 3,48* | 6.83*,** | 5.27*,** | 3.11*,** | 3.18*,** | 6.86* | 4.95* |
| **Noah** | 3.58*,** | 4.20*,** | 6.97* | 5.98* | 3.49*,** | 3.78* | 6.93* | 5.10* |

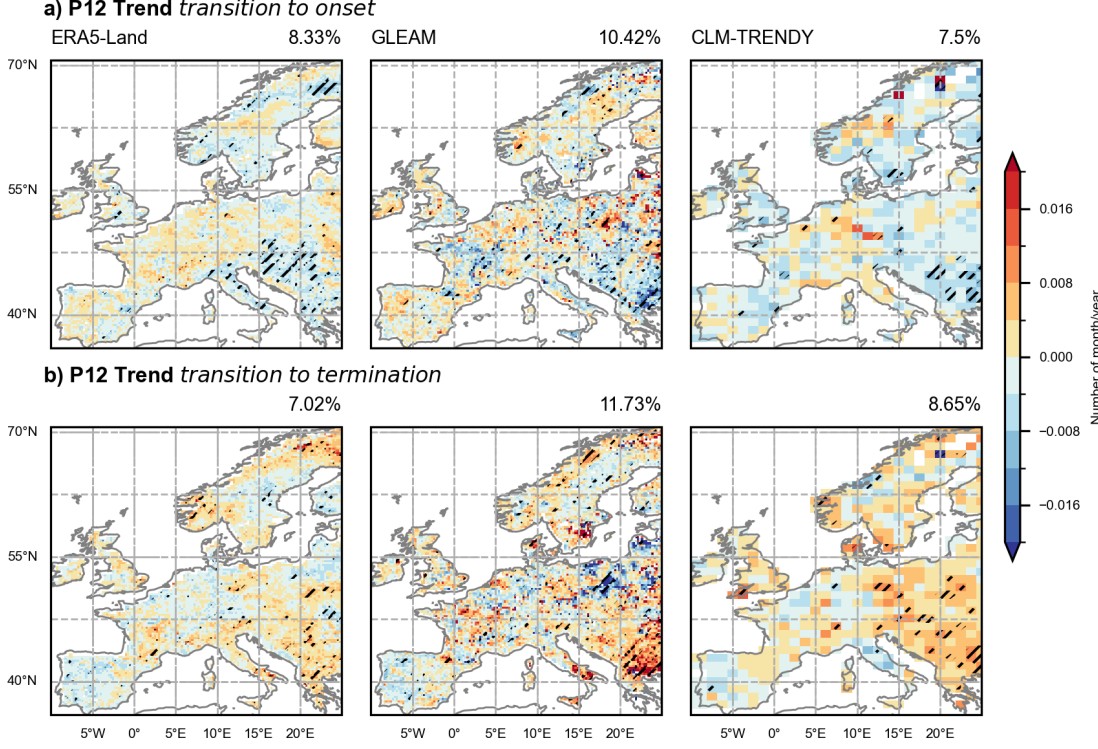

**Figure 5.** Linear trend coefficients in the transition periods to a) DO and b) DT during 1980–2020 in the P12-based definition. The regions where the trend coefficients are statistically significant at a 95% confidence level are dashed. The percentages in the upper right indicate the areas where the trends are statistically significant.



We did not observe the same spatial patterns of trends in the SM3-based DO&T (Supplementary Figure S4). The trends

in SM3-based duration show a more sporadic spatial distribution in all datasets, indicating that the changes in soil moisture related to DO&T are not as evident as those in precipitation.

## 4.2 Preferred seasons for drought onset and termination

Regarding the preferred seasons for DO&T, the ratios of the number of DO for each season to the total number of DO during 1980–2020 (Section 3.4) are calculated for ERA5-Land, GLEAM, and CLM-TRENDY. Fig. 6 shows the median of the

310 frequencies of DO&T across each subregion in Fig. 1. The values over the entire Europe are included in the supplementary Figures S5 and S6.

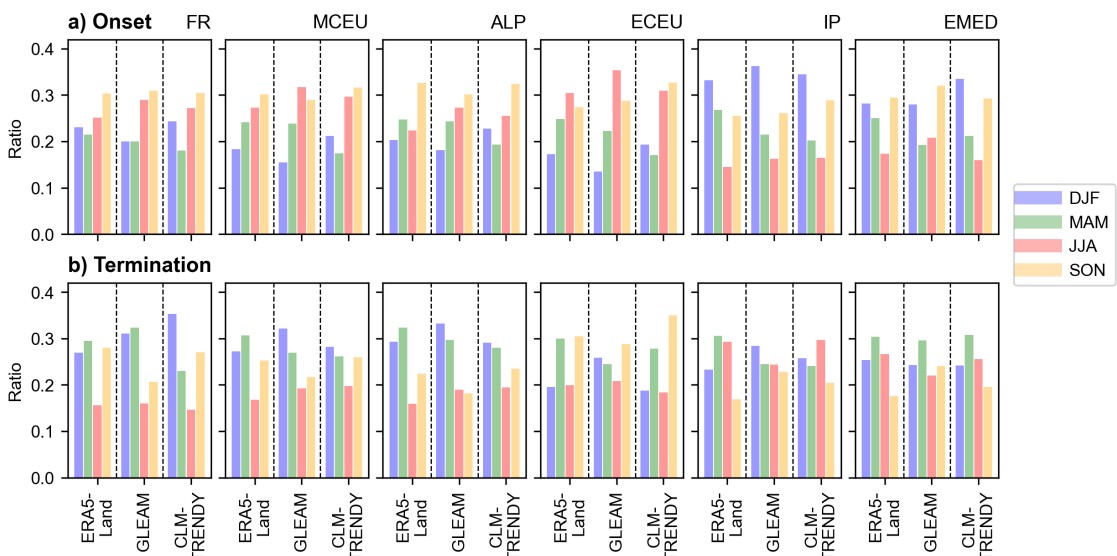

**Figure 6.** a) The medians of occurrence ratio of onsets over all grid points for each season, dataset, and domain from Fig. 1. (b) is the same as (a) but for terminations.

For DO, Fig. 6a shows that although more subregions are considered, the seasonality of DO is similar across the regions of CEU (FR, MCEU, ALP, and ECEU). All three datasets indicate SON and JJA as the most likely seasons for DO, while the most unlikely seasons vary between DJF and MAM, depending on the dataset. In IP, DJF is the most likely season for DO.

The second most likely season varies across the dataset: ERA5-Land presents MAM, and GLEAM and CLM-TRENDY denote SON. However, the difference in the frequency ratio between SON and MAM in ERA5-Land and GLEAM is relatively small. Over EMED, DJF and SON are the most likely seasons for DO. In general, over the entire MED (IP and EMED), the most unlikely season for DO is JJA, except for GLEAM in EMED. In EMED, GLEAM denotes MAM as the most unlikely season, followed by JJA. However, the difference in the frequency ratio between the two seasons is again small.



For DT (Fig. 6b), western central Europe (WCEU; consisting of FR, MCEU and ALP) shows that DJF and MAM are the most frequent seasons for DT. In ECEU, the most frequent seasons for DT are SON and MAM in ERA5-Land and CLM-TRENDY, and SON and DJF in GLEAM. MAM is the third most frequent season in GLEAM, with only a small difference in frequency ratio compared to DJF. In CEU (FR, MCEU, ALP, and ECEU), the least likely season for termination is JJA. Over IP, the frequency ratios of DT are relatively similar across the four seasons, with the values significantly depending on

the dataset. In EMED, MAM is the most likely season for DT, followed by either DJF or JJA, depending on the dataset. In the end, it is notable that the most likely seasons for DT are less consistent across the datasets and regions compared to DO.

    In general, the preferred seasons for onsets coincide with each region's wettest seasons. In CEU (FR, MCEU, ALP, and ECEU), these periods are JJA followed by SON, and in MED (IP and EMED), these seasons correspond to SON and DJF, as indicated by their annual cycles, shown in Fig. 7. Onsets occur less frequently during the dry seasons, which are DJF and

MAM in CEU, and MAM and JJA in MED. This result highlights the crucial role of precipitation availability and, therefore, potentially related circulation patterns occurring during the wet seasons in initiating dry conditions leading to DO.

    In the case of termination in CEU, it generally occurs more frequently in all seasons except in JJA, which is the wettest season in the region. In MED, there is no clear temporal consistency across the datasets. This finding indicates that termination is less seasonally constrained, potentially indicating varying factors that can end droughts, which can be less apparent than

those that induce drought onset. Nevertheless, it is important to note that while there are some preferred seasons for DO&T exhibiting higher frequency ratios, DO&T can still occur in other seasons, as shown in Fig. 6. The frequency ratio of DO&T during the most likely seasons does not overly exceed the ratio during other seasons. For example, the frequency ratios of DO&T in the first three most likely seasons generally exceed 0.2 across the majority of regions and datasets.

### 4.3   Large-scale circulation modes associated with drought onset and termination

We examine the contributions of four large-scale circulation modes — SCA, NAO, EA, and EA-WR — to DO&T by calculating the mean values and frequencies of the corresponding modes during the DO&T transition periods. Since Fig. 6 shows some spatial consistency in the seasons of frequent DO&T occurrences across WCEU, we perform the analysis by considering WCEU as a single region. For the transition periods, we use those based on P12. This is because the transition periods for DO&T based on P12 are relatively longer (Figs. 3 and 4); therefore, we can assess lagged and persistent influence of circulation

modes on leading droughts instead of their instantaneous effects on soil moisture. Prior to calculating the frequencies of the climate modes involved in the transition periods to DO&T, the relationship between each of the four circulation modes and seasonal precipitation variability is evaluated using the Pearson correlation analysis (Fig. 8).

    Fig. 8 shows the correlation coefficients between seasonal mean precipitation and each circulation mode during DJF and JJA. Other seasons are not shown here as they exhibit transitional spatial patterns between these two seasons (for those interested,

they are included in the supplementary Figure S7). The influences of these circulation modes on precipitation in Europe have been investigated in many previous studies (e.g., those mentioned in Section 2.2). Therefore, here we simply describe the signs of the influences of each mode in affecting precipitation in different regions and seasons to later connect these relationships to the modes occurring during DO&T.



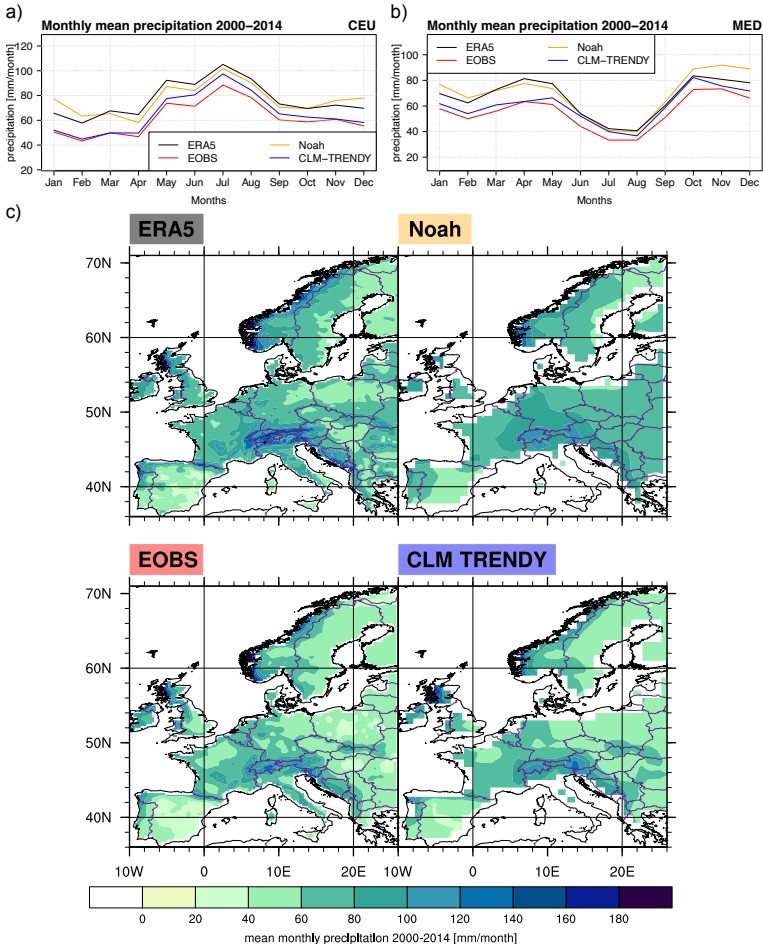

**Figure 7.** Annual cycles of precipitation during the reference period 2000–2014 from ERA5-Land, Noah forcing, E-OBS, and CLM-TRENDY forcing for a) CEU and b) MED. (c) Annual mean precipitation during the same period over Europe.

In DJF, SCA is positively correlated with precipitation over MED (IP and EMED), while it is negatively correlated with pre-
cipitation over CEU. In JJA, the signs of the coefficients remain positive in MED. However, the regions where the coefficients are statistically significant are reduced, primarily over the Balkan region. The positive coefficients are expanded to CEU, but show statistically significant values only over ECEU and western France.

For NAO, a clear seasonal inversion in the correlation coefficients is observed. In DJF, statistically significant negative correlation coefficients are dominant over IP and in northern MED. In contrast, statistically significant positive coefficients are
apparent in central and northern Europe. The signs of the correlation coefficients are reverted during JJA, with northern Europe and CEU showing negative correlation coefficients and southern regions presenting positive coefficients, although not many are statistically significant.



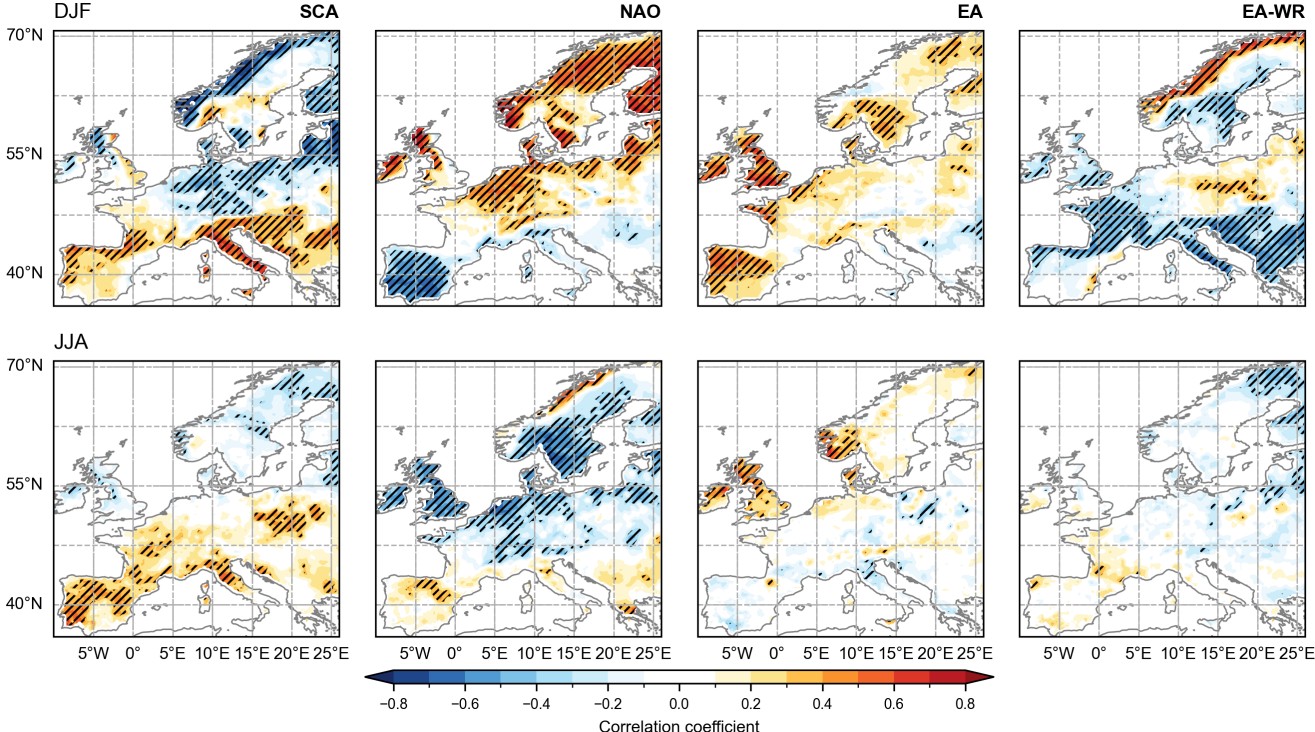

**Figure 8.** Pearson correlation coefficients between seasonal mean indices of circulation modes and seasonal precipitation for DJF (upper panels, and JJA (lower panels). The regions where the coefficients are statistically significant at a 95% confidence level are dashed.

The influences of EA and EA-WR over CEU and MED are more apparent in DJF than in JJA. EA is correlated positively and statistically significant over IP, northern France and west Alpine regions. It is negatively and statistically significantly correlated over small areas in the eastern Mediterranean. For EA-WR, regions with significant negative correlations are found over southern Europe, except the central and southern parts of IP and western France. Over ECEU, there are regions with light positive correlation coefficients, which are statistically significant. In JJA, both circulation modes do not exhibit many regions over MED and CEU where the correlation coefficients are statistically significant.

Table 3 provides a summary of the circulation modes and their phases (either positive or negative. Red or blue label, respectively) that occur more frequently during specific drought phases (DO and DT) for each season. Only the circulation modes with mean values during a drought phase that significantly differ from their climatological mean are shown. Additionally, the circulation modes that change the phase from DO to DT are highlighted in yellow, as such a shift can serve as an early DO&T prediction. A change in the phase from DO to DT suggests that the circulation mode might play a role in initiating or terminating droughts by driving an opposite circulation condition. The numerical values corresponding to the table can be found in the supplementary Table S1.





**Table 3.** Circulation modes and their phase (either positive or negative) that occur more frequently during DO and DT for each season and subregion. Those modes that change their phase sign from DO to DT are highlighted in yellow. The corresponding mean values and the frequencies can be found in Table S1 in the supplement.

|     | WCEU | ECEU | IP | EMED |
|-----|------|------|-----|------|
| **DJF** | SCA (DO – DT) | SCA (DO – DT) |  |  |
|  | NAO (DO – DT) | NAO (DO – DT) | NAO (DO – DT) | NAO (DO – DT) |
|  |  |  | EA (DO – DT) | EA (DO – DT) |
|  | EAWR (DO – DT) | EAWR (DO – DT) | EAWR (DO – DT) | EAWR (DO – DT) |
| **MAM** | SCA (DO – DT) |  | SCA (DO – DT) |  |
|  | NAO (DO – DT) | NAO (DO – DT) | NAO (DO – DT) | NAO (DO – DT) |
|  |  |  | EA (DO) |  |
|  | EAWR (DO – DT) | EAWR (DO – DT) |  | EAWR (DO – DT) |
| **JJA** | SCA (DO – DT) | SCA (DO – DT) | SCA (DO – DT) | SCA (DO – DT) |
|  |  |  | NAO (DO – DT) | NAO (DO – DT) |
|  | EA (DO – DT) | EA (DO – DT) | EA (DO – DT) |  |
|  |  |  | EAWR (DO – DT) |  |
| **SON** | SCA (DO – DT) | SCA (DO – DT) | SCA (DO – DT) |  |
|  | NAO (DO – DT) | NAO (DO – DT) | NAO (DO – DT) | NAO (DO – DT) |
|  |  | EA (DO – DT) |  | EA (DO – DT) |
|  | EAWR (DO – DT) | EAWR (DO – DT) |  | EAWR (DO – DT) |

SCA is observed across all seasons in WCEU, although the dominant phase (positive or negative) during DO&T varies by season. In JJA over all regions (WCEU, ECEU, IP, and EMED), a negative SCA occurs more often during DO, and a positive phase occurs more during DT, which corresponds to the expanded influence of SCA to CEU during that season (Fig. 8). Over IP, SCA appears to occur also frequently during MAM and SON, with the same phase characteristics. SCA is also dominant in WCEU but with the reverted condition with the positive phase during DO and negative during DT, which again seems to be in line with Fig. 8, with negative correlation coefficients between SCA and precipitation over CEU.

NAO shows significant influence across nearly all regions and seasons, but the frequent phases vary greatly among regions. In WCEU and ECEU during DJF and MAM, positive NAO is observed more frequently during both DO&T. In SON, positive NAO appears more during DO, while negative NAO is more common during DT. The NAO occurrence is particularly evident in the IP region. During DJF, MAM, and SON, positive NAO occurs more frequently during DO, whereas negative NAO is more common during DT. The opposite conditions occur in JJA. This result is in line with Fig. 8, which shows the opposite seasonal influence of NAO on precipitation. In EMED, similar values of the NAO frequencies to those of IP are observed in DJF and JJA, while in other seasons, positive NAO conditions dominate during DO&T.

The influence of EA is more limited and noticeable only in certain seasons, also shown as Fig. 8. In WCEU, positive EA is more frequent during both DO&T. The same happens in ECEU during JJA, but in SON, EA is more positive during DO and





negative during DT. In IP, EA influences DO&T in DJF and JJA, but with opposite phases: negative EA appears frequently during DO, with no significant influence in DT. In EMED, EA becomes more apparent in DJF and SON, with more frequent positive conditions during DO and negative ones during DT.

For EA-WR, the most influential regions are WCEU, ECEU, and EMED during DJF, MAM, and SON, all regions with similar characteristics. Across all seasons and regions, positive EA-WR occurs more frequently during DO, while negative EA-WR is more common during DT. However, this relationship is less clear over IP. In IP, positive EA-WR is observed during DO and negative during DT in DJF, whereas both phases show negative EA-WR during JJA. This weaker influence of EA-WR over IP is consistent with Fig. 8. Statistical significance is weak in JJA, which is also shown in Fig. 8.

Overall, the result indicates that the dominant patterns that could potentially serve as predictors vary by region and season. In WCEU, these modes are SCA (with opposite phase conditions in DT&O between DJF and JJA), EA-WR and NAO (only in JJA). For ECEU, these are EA-WR (DJF, MAM and SON), SCA (JJA) and NAO (SON). For IP, NAO is apparent for all seasons, and the other three modes vary depending on the seasons. FOR EMED, EA-WR is evident in all seasons except JJA, but the other modes vary from one season to another.

## 5 Discussion

### 5.1 Difference in the duration and the occurrence timing of drought onset and termination among the dataset

Our study reveals the transition period to drought onset (DO) and termination (DT) is sensitive to how it is defined. The P12-based definition of the transition period shows a longer duration for DO than for DT, whereas the SM3-based definition exhibits relatively small differences between DO and DT. A longer transition period to DO means that more cumulative P deficit is required to initiate DO than cumulative positive P anomalies needed for DT. This finding is consistent with some previous studies by Mo (2011) and Řehoř et al. (2021), which analyzed drought onset and demise using cumulative precipitation of different time scales. (Mo, 2011) examined transition periods to drought onsets and termination for the U.S. using the monthly P anomalies, while Řehoř et al. (2021) used daily P for central Europe. The longer transition periods to DO observed in their studies, as well as ours, suggest that DO is more predictable than DT. The difference between DO and DT is not evident in the SM3-based transition definition. For most of the regions and datasets, the SM3-based DT is slightly longer than DO. This discrepancy between the P-based and the SM-based definitions was also remarked by Seager et al. (2019), a finding which further emphasizes the dependence of transition to DO&T duration on how the transition period is defined.

Another finding of this study is the dependence of the DO&T characteristics on the soil moisture dataset. The duration of the transition periods, as well as the seasonal frequency of DO&T, vary across the datasets. We observed much shorter DO&T transition periods in ERA5, while GLEAM presents longer values. The values from other datasets fall within the range of these two datasets. We analyzed the potential origins of this discrepancy but did not identify specific influences of P or evapotranspiration. P, which is the primary input forcing for LSMs, exhibits relatively good spatial and temporal consistency across the datasets (Fig. 7). For evapotranspiration, although the datasets present some spatial differences in representing this variable, there is no significant relationship between the magnitudes of evapotranspiration and the duration of DO&T.



Therefore, we concluded that the observed discrepancies in the transition periods and frequencies of DO&T among the
datasets are primarily explained by LSMs' internal physics that determine the final water balance in soil moisture (Fang et al.,
2016; Berg and Sheffield, 2018). This result highlights the need for caution when conducting drought studies based on a
single soil moisture dataset. Nevertheless, we observed the existence of some common seasons of DO&T occurrence, which
implies a certain range of similarity in the temporal variability of soil moisture across different LSMs. Overall, the finding that
the characteristics of DO and DT depend on the definitions and datasets used seems to imply that these choices will impact
decision-making related to droughts.

We focus on the climatology of DO&T, expanding the climatology of European droughts given by Lloyd-Hughes Benjamin
and Saunders Mark A. (2002). Until now, while DO have been relatively more studied due to practical reasons, spatiotemporal
characteristics of DT have been a topic that has attracted less attention. We found that DO occurs more frequently during the
wet seasons, while DT occurrence is seasonally not constrained in Europe. This may also explain the observed discrepancy
in the seasonality of termination across the datasets. The fact that DT is seasonally not constrained agrees with the study of
Gibson et al. (2022), which examined meteorological DO&T based on SPI in Australia. Studies that investigated terminating
drivers of different types of droughts — meteorological, soil, and hydrological — indicate diverse drivers of different origins
and time scales. These include large-scale climate modes, similarly as studied here (e.g., Parry et al., 2016; Shah and Mishra,
2020; Gibson et al., 2022), or synoptic weather events such as tropical cyclones (Lam et al., 2012), extratropical cyclones
(Stojanovic et al., 2021), and frontal systems (Maxwell et al., 2017). Synoptic scale events have a much shorter time scale
and can lead to a fast termination of droughts. These varied drivers that act on different time scales can be the reason why
terminations are seasonally less constrained than onset.

## 5.2 Trends in the transition periods to drought onset and termination

Warmer climate is expected to cause significant changes in temporal characteristics of droughts (Trenberth, 2011; Yuan et al.,
2023; Walker and Van Loon, 2023). For instance, flash droughts, droughts characterized by a rapid onset, have increased across
the globe since 1950 (Yuan et al., 2023). The increased frequency of flash droughts indicates an acceleration in drought onset
speed. These changes are attributed to anthropogenic climate change driven by a decrease in precipitation and an increase
in evapotranspiration, the latter causing a fast depletion of soil moisture. The regions experiencing a noticeable increase in
flash droughts are the global humid areas, including Europe. Additionally, anthropogenic activities that alter water storage,
especially those affecting stream flow, are another driver that affects drought characteristics and feedback within droughts
(Van Loon et al., 2016a; Margariti et al., 2019). Margariti et al. (2019) demonstrated extended drought termination rates due to
human influences on water storage in Europe.

Our result, based on three soil moisture and precipitation datasets, does not show robust changes in the trends of DO&T
transition periods during 1980–2020 over most of Europe. Significant trends are only observed in the eastern Mediterranean,
showing decreased transition periods to DO and increased transition periods to DT. The consistency of trend patterns across
all three soil moisture datasets over the same regions supports the robustness of this result. This finding seems to be in line
with the above-mentioned studies on the changes in temporal characteristics of droughts, indicating that DO has become faster.



However, changes in DO&T are only apparent in the precipitation-based DO&T, but not in the soil moisture-based definition. This discrepancy between the definitions, as well as the limited regions with significant trend changes, could be related to the relatively short 41-year analysis period. We use these 41 years as it is the extent of available soil moisture data for the majority of the datasets.

Fast DO signifies that the timing of drought initiation becomes more difficult to predict, while slow DT implies prolonged droughts and extended periods of economic suffering. Therefore, this result supports more understanding of present and future DO&T is needed to prepare efficient adaptation and mitigation strategies to manage drought impacts.

## 5.3 Circulation modes as a potential early warning for drought onset and termination

Large-scale atmospheric circulation modes influence DO&T across Europe, with their effects varying by region and season. The modes that change their phase from DO to DT can serve as early warnings for DO&T as they provide opposite circulation conditions to initiate and terminate droughts. However, even when a circulation mode does not shift its phase from DO to DT, it does not mean that they have no influence on DO&T. They can be more influential on only one drought phase, or joint effects with other circulation patterns can be more important. We found that those modes that can serve as early warnings vary with the regions and seasons, and sometimes the same mode presents a reversed-phase condition between two seasons, for instance, SCA over WCEU and NAO over IP. In WCEU, SCA is positive for DO and negative for DT, but this is the opposite in JJA, with negative SCA for DO and positive SCA for DT. This is the same in IP with NAO during DJF and JJA.

Our analysis, in which we examine the influences of circulation patterns on DO&T by considering the circulation conditions during the transition periods to DO&T, follows a similar approach to Almendra-Martín et al. (2022) where lagged responses of soil moisture to some atmospheric patterns (NAO, AO, and El Nino-Southern Oscillation) are investigated. In our case, we only included circulation patterns that are known to influence the climate of Europe and also considered seasonal analysis. The relationship between NAO and DO&T is comparable to the finding of Almendra-Martín et al. (2022) showing that the negative lagged effects of NAO are dominant over almost the entire Europe. The difference is that our result indicates that the effects of NAO are pronounced over the Iberian Peninsula. We also found that, unlike other patterns that influence a limited region, EA-WR in DJF is influential on DO&T over all CEU and MED.

To assess these relationships, we used a simple method by calculating frequencies and evaluating the mean values with t-tests for each pattern, season, and drought phase. This is because many sophisticated statistical methods with multivariate approaches require statistical independence among the variables, which is, in most cases, difficult to achieve. Additionally, while achieving this independence, the joint influences of the variables on droughts can be obscured. Although we applied a relatively simple method, it still provides a straightforward identification of dominant circulation modes, which is also consistent with previous literature on large-scale influences on droughts and precipitation over Europe (e.g., Vicente-Serrano et al., 2011; Kingston et al., 2015; Markonis et al., 2018). Further analysis will still be necessary to assess the combined influence of multiple variables on soil moisture droughts.





# 6 Conclusion

We have examined temporal characteristics, namely the typical duration and the seasons of occurrence, and large-scale circulation modes leading to drought onset and termination (DO&T) in central (CEU) and Mediterranean Europe (MED) from 1980 to 2020, using five different soil moisture datasets: ERA5-Land, GLEAM, SoMo, Noah, and CLM-TRENDY. We use two different ways of defining transition periods to DO&T: one is based on precipitation, and another one is based on the same soil moisture drought indices. Our main findings are:

- The temporal characteristics of DO&T depend on the definition of transition periods and the datasets. The observed discrepancies in the transition periods and frequencies of DO&T among the datasets seem to be primarily related to LSMs' internal physics that determines the final water balance in soil moisture. This finding implies that these choices will impact decision-making related to droughts.

- In all datasets, when considering the P12-based transition periods, DO is longer than DT, indicating potential predictability of onset. However, this may change for some regions in Europe in the future, as there is an indication that DO has become faster while DT has shifted to be slower.

- Regarding the seasonality of DO&T, the wet seasons are the most likely periods for onset in CEU and MED. These are summer (JJA) and autumn (SON) for CEU, and autumn (SON) and winter (DJF) for MED. This emphasizes the importance of precipitation availability and related atmospheric circulation during the wet seasons in initiating droughts. Nevertheless, although there are some preferred seasons for onsets, they can still take place in other seasons, and the frequency of occurrence during the most likely seasons does not overly exceed the frequency during other seasons.

- For termination, there are no consistent seasons of occurrence among the datasets, and the datasets show more discrepancies than those of onset. Circulation modes that are associated with DO&T are identified, showing those that can serve as early warning for some regions and certain seasons. For instance, the impacts of NAO on drought O&T are clear in all seasons in IP, while for other regions, EA-WR is more influential on drought O&T except in summer.

Our study is one of the few that have investigated the onset and termination of soil moisture droughts in Europe, providing clear, distinct temporal and circulation characteristics involved in each drought phase. This study that provides climatology of DO&T for the present day can serve as a base to assess future changes in soil moisture DO&T. A downside of this study would be that the temporal extent of the datasets, which is from 20 to 40 years, may not be long enough to capture all possible variability of DO&T and the temporal trends.

It is also important to remark that regional differences in the temporal characteristics of DO&T are observed within CEU and MED. Europe is characterized by complex topography and is located in the transition zone from a semi-arid climate in the south to a relatively temperate wet climate in the north. Therefore, each climate may present different drought characteristics, which are related also to the seasonality of precipitation. Our analysis separates regional domains considering these climate aspects largely following Christensen and Christensen (2007). However, it may not take into account very small-scale regional differences that occur within these domains.



More understanding of drought onset and termination is crucial to improve early predictions and preparedness for potentially devastating droughts, especially when drought characteristics are expected to change due to global warming. More knowledge on drought onset and termination — especially on termination, which has been more disregarded — can provide valuable insight for water management, drought adaption and mitigation planning. Investigating how the life cycle of droughts is affected by anthropogenic warming will be the follow-up research question to anticipate how these extreme hydrological events may evolve in the future.

*Code availability.* The python script with a function for drought phases estimation is available on the corresponding author's GitHub https://github.com/wmk21/drought_estimation/.

*Data availability.* The datasets used in this study are available online in: ERA5 and ERA5-Land at https://cds.climate.copernicus.eu/, GLDAS LSMs at https://ldas.gsfc.nasa.gov/gldas, GLEAM at https://www.gleam.eu/, E-OBS at https://www.ecad.eu/download/ensembles/download.php, and SoMo.ml at https://www.bgc-jena.mpg.de/geodb/BGI/somo_ml_v1.php. The monthly NAO index is obtained from https://climatedataguide.ucar.edu/climate-data/hurrell-north-atlantic-oscillation-nao-index-station-based.

*Author contributions.* WMK designed the study, performed the principal analysis, and drafted the manuscript. SJGR provided the analysis of the precipitation cycles and feedback on the methodology and results. All the authors contributed to the writing of the manuscript.

*Competing interests.* The authors declare that they have no conflict of interest.

*Acknowledgements.* We thank all the research groups that produced the datasets used in this study and for making their output publicly available. We acknowledge the Copernicus program for the ERA5 and ERA5-Land data (Hersbach et al., 2020; Muñoz-Sabater et al., 2021) available in Copernicus Climate Change Service Climate Data Store, the NASA/NOAA Global Land Data Assimilation System for the Noah and Catchment Land Surface Model datasets (Rodell et al., 2004), https://www.gleam.eu/ for GLEAM v3 (produced by Dr. Akash Koppa and validated by Dr. Petra Hulsman, Martens et al., 2017), ECA&D for E-OBS (Klein Tank et al., 2002; Klok and Klein Tank, 2009; Cornes et al., 2018), O and Orth (2021) and Max Plank Institute for Biogeochemistry for SoMo.ml dataset, and the Climate Data guide project at NCAR for the NAO index (Schneider et al., 2013; Hurrell et al., 2023), and the Global Carbon Project (Friedlingstein et al., 2022) for the CLM-TRENDY simulation. WMK thanks Dr. Vít Svoboda (JILA, CU Boulder) for his comments on plotting and the drought estimation function, Dr. Isla Simpson (CGD, NCAR) for feedback during the first draft, and Dr. Daniel Kennedy (CGD, NCAR) for providing CLM-TRENDY datasets. WMK acknowledges funding from the Swiss National Science Foundation (grant number P500PN_206653). This work is supported by the NSF National Center for Atmospheric Research, which is a major facility sponsored by the National Science Foundation under the Cooperative Agreement 1852977.



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
