# Peer review of "Characterizing Soil Moisture Drought Onset and Termination in Europe"

_EGUsphere, 2025_

## Referee Comment (RC2)

**REVIEW: Characterizing Soil Moisture Drought Onset and Termination in Europe**

I would like to congratulate the authors for such a well-conducted and insightful characterization of droughts, their evolution in time, discrepancies among datasets, and the role of large-scale atmospheric modes of variability. The study clearly highlights the sensitivity of drought analyses to dataset choice and suggests that the evolution of these extremes may be predictable through persistent large-scale modes of variability. Although the manuscript is somewhat lengthy, it is well written, clearly structured, and supported by high-quality figures.

Nevertheless, I have some major concerns that should be addressed before the manuscript can be considered for publication. The comments below are provided constructively to help improve the quality and impact of the study.

**Major comments:**

**Structure of the paper:**

The section 2.1 introducing the soil moisture datasets is overly long and excessively detailed. The authors should focus on essential and objective information, such as the datasets used, their main differences, spatial and temporal resolutions, and the analysis period. Additional complementary details (Table 1), while relevant, are not strictly necessary for the analysis and could be moved to the Supplementary Material. Consider this as a suggestion intended to help reduce the overall length of the manuscript.

**Datasets**

A new version of GLEAM (GLEAM4.2) has been made available and with important updates relative to GLEAM version 3. Authors should consider using this updated version and recomputing the results accordingly.

**Methods:**

There is some lack of consistency and clear information on the time period and datasets chosen for the analysis. For example, in Section 4.1, part of the analysis is conducted using all available datasets but considering two different time periods (2000–2019 and 2000–2020), whereas the last section presents results based on only three datasets and over an extended period (1980–2020). I know the authors struggled to find a common temporal coverage among the six datasets. Nevertheless, I would prefer to define a single and longer time period with a consistent spatial resolution among datasets, so that a coherent and robust analysis is employed (even if this requires reducing the number of datasets considered).

I find little justification for prioritizing the presentation of transition periods to drought onset and termination based on precipitation, when drought identification and the corresponding DO and DT are defined entirely using soil moisture—a variable that responds partially to precipitation. While I understand that the results may appear more "appealing" from a statistical perspective (due to a larger number of grid points showing significant trends in Figure5), unless there is a strong and proper justification (which is not presented by

the authors), conceptually, I find no clear rationale for defining the onset and termination of a soil moisture deficit while neglecting the role of temperature, land cover, and soil characteristics. This issue is particularly relevant given the strong spatial variability in soil moisture sensitivity to precipitation and temperature, as well as regional differences in the relative contribution of precipitation.

It would be interesting to assess the sensitivity of the DO&T transition periods to the drought duration. Basically, is the time required to end a drought proportional to the duration of the event itself? If so, could this be linked to changes in land cover and soil properties (porosity/ hydrophobic)? This could be explored by conducting the analysis for distinct groups of droughts categorized by event duration.

I've found some misleading and contradictory descriptions of the drought onset and termination concepts that are undermining the clarity of the methods. In lines 204-205, it is unclear what the authors mean by "after a continuous period of positive SM3". This description does not align to what is shown in Figure 2a, where the onset is marked as the first month when SM3 drops to values below -1σ, following a sequence of months with SM3> -1σ (including positive and negative values). Furthermore, in line 205, the authors mention the following: "On the contrary, a drought termination (DT) is the first monthly time step that SM3 reaches above -1σ from the minimum SM3". However, this raises an important question: if SM3 drops below −1σ again in the subsequent month, is it still appropriate to consider the drought as terminated?
Finally, how sensitive are the results to the choice of the SM threshold? Do the spatial patterns and inter-dataset differences change substantially when SM thresholds other than −1σ are considered?

**Analysis of the results:**

Regarding the analysis focused on the impact of the several modes of variability on DO&T, the authors tend to explain this interlink mainly via precipitation anomalies. Knowing that DO&T are identified through soil moisture anomalies and that precipitation deficits explain only partially fluctuation in soil water content, I feel like the analysis is missing other key aspects. Similarly to Fig. 8, it would be good to show the spatial distribution of the correlation coefficients between the modes of variability and temperature + VPD (perhaps as Supplementary Material). Information about these variables would complement the explanation for the causal link between atmospheric dynamics and soil moisture variability.

**Minor Comments:**

**Line 55:** Change accordingly: "Due to the complex and multifaceted nature of droughts (Cook et., 2018), determining the precise timing of initiation and termination of droughts is challenging, which may explain the lack of studies addressing this topic.

**Line 60:** Change accordingly: "(...) are typically slowly-evolving events, with impacts on ecosystems becoming mainly noticeable in latter stages, after progressed accumulated periods of precipitation and/or soil moisture deficits (...)"

**Line 62:** It is unclear what is meant by "dry" or wet conditions". In this context, this terminology is vague and lacks scientific rigor.

**Line 74:** " Parry et al. (2016) reviewed drought terminations in the British Isles (..)". It reads a bit weird. Please rephrase it.

**Line 110:** Remove the following: "(...) and many solely observation-based soil moisture datasets are generally short and not continuous in time and space".

**Line 120:** What do the authors mean by "present"? For instance, GLEAM4.2a only goes until December 2023 Anyway, this information is unnecessary. The important thing is to mention what was the time period defined to conduct the analysis which is something I don't find mentioned explicitly in the manuscript.

**Line 126:** Change accordingly to keep consistency: "The spatial resolution of SoMo is 0.25° × 0.25°:

**Line 130:** Remove the following: "The output from GLDAS version 2.1 is used taking the period 2000–2020".

**Line 140:** GLEAM4 uses precipitation from MSWEP v2.8 (https://www.nature.com/articles/s41597-025-04610-y) that can be downloaded here (https://www.gloh2o.org/)

**Line 147:** "Modes of circulation patterns". Typically in the literature these are referred to as "Modes of atmospheric circulation variability". Please due the necessary changes over the whole manuscript.

**Line 151:** Change accordingly: The NAO index was retrieved from the NCAR climate data guide (https://climatedataguide.ucar.edu/type/climate-indices/circulation/nao), while the other indexes were obtained from KNMI climate explorer (https://climexp.knmi.nl/).

**Line 153-156:** "the index is calculated as the leading modes of Empirical Orthogonal Function (EOF)". This lacks scientific rigor. If I got it right the authors used the PC-based NAO index (Hurrel - NAO) which relies on the PC1 obtained from decomposing the SLP spatiotemporal field using a principal component analysis. The leading mode of the EOF gives the main spatial pattern of SLP variability, while the true "NAO index" is obtained from its time series (PC1). Check also the lines 161-162.

**Line 165:** Change accordingly. "We considered Europe as our region of study, particularly the central (CEU) and Mediterranean Europe (MED)"

**Lines 173-174:** Change accordingly: Surface soil moisture levels over a layer of 10cm depth were used. While GLEAM, SoMo, Noah, and CLM-TRENDY provide information directly for this layer, the corresponding value for ERA5-Land is obtained by estimating how (...)"

**Line 192:** Do you mean something like: "(...) where an individual time step $t$ considers the mean soil moisture conditions over the previous two months, from t-2 to t0. Why was a

three-month time window chosen? Was a sensitivity analysis performed using different time windows, and if so, how sensitive are the results to this choice?

**Lines 197-198:** Rephrase it please

**Lines 200-201:** "A drought is a period when SM3 stays below one negative standard deviation (-1σ)". By "period," do you mean one month, multiple months, and must the months be consecutive?

"A threshold of -1σ in a standardized drought index corresponds approximately to the 15.9th percentile level". Considering that SM3 was computed pixel-wise, how was exactly this value derived? Can you clarify?

**Lines 221-222:** "The 12-month scale is chosen because it is the typical time scale of drought indices". This is misleading... The scale used truly depends on what type of drought you are interested in identifying (meteorological, hydrological, agricultural). Also the reference Mo, 2011 does not say anything regarding this issue.

**Lines 250-251:** How do the authors plan to count the number of occurrences of a mode of variability? The mode is always there, the only things that change are either the signal or the intensity.

**Line 258:** An equal time period should be used in order to ensure that any differences in the results are fully explained by biases across datasets.

**Line 261:** I know what P-12 stands for... However this acronym is not defined in the manuscript.

**Line 278-281:** I understand the authors' point that certain regions exhibit longer drought onset and termination (DO&T) periods, consistently across datasets, drought phases, and transition definitions. However, this idea should be explained more clearly, and the sentence should be rewritten for better English and readability.

"In general, CEU tends to show slightly longer DO&T duration than MED, in which this difference is more pronounced in the DT". This reads weird

**Line 292:** Is really the "study period" 1980-2020? I see no consistency here. So far the analysis was conducted for the periods 2000-2019 and 2000-2020...

**Figure 5.** I suggest depicting the statistically significant values with some kind of hatches (dots), contours or shades. The way it is represented right now makes it hard to identify which ones are really significant.

**Figure 7.** I would move this figure to Supp. Material. Although important, it shows results that are only a complement to the analysis and are not pivotal. Please take into account that this is just a suggestion aiming to shorten a bit the size of the manuscript.

**Lines 408-409:** " A longer transition period to DO means that more cumulative P deficit is required to initiate DO than cumulative positive P anomalies needed for DT". Poor English. Please rephrase it.

**Line 412-413:** "The longer transition periods to DO observed in their studies, as well as ours, suggest that DO is more predictable than DT". What is the causality between both things? Because it's easier to predict large-scale modes of variability than other transient eddies? This needs to be explored a bit more.

**Line 420-421:** "We analyzed the potential origins of this discrepancy but did not identify specific influences of P or evapotranspiration." Maybe lower high-frequency (daily/weekly) P and E variability in ERA5?

**Line 436-442:** Agree. I suspect that drought termination (DT) is more sensitive to short-lived precipitation events and, consequently, highly influenced by transient eddy systems, which are less constrained by seasonal patterns (?).